

# Flotillin-1 promotes EMT of gastric cancer *via* stabilizing Snail

Ying Huang[1], Yun Guo[1], Yi Xu[1], Fei Liu[2] and Suli Dai[2]

[1] The Fifth Hospital of Shijiazhuang, Shijiazhuang, Hebei, China
[2] Research Center, The Fourth Hospital of Hebei Medical University, Shijiazhuang, Hebei, China

## ABSTRACT

Gastric cancer is one of the most common malignancies worldwide and has been identified as the third leading cause of cancer-related mortality. Flotillin-1 is a lipid raft-associated scaffolding protein and plays an important role in the progression and development of several malignant carcinomas. Flotillin-1 is involved in epithelial-mesenchymal transition (EMT) process of several solid tumors to promote metastasis. However, the detailed characteristics and mechanisms of Flotillin-1 in gastric cancer have rarely been investigated. In this study, we found Flotillin-1 upregulated in gastric cancer, and the high expression of Flotillin-1 correlated with a worse prognosis. The migration and invasion ability of gastric cancer cells was upregulated by overexpressing Flotillin-1. Knockdown of Flotillin-1 inhibits gastric cancer cells metastasis. Flotillin-1 is a key regulator of EMT process and promotes gastric cancer cells metastasis through inducing EMT. Flotillin-1 may interact with a deubiquitinase to inhibit the ubiquitination of Snail in gastric cancer cells to promote EMT process. Our study provides a rationale and potential target for the treatment of gastric cancer.

## INTRODUCTION

Gastric cancer is one of the most common malignancies worldwide and has been identified as the third leading cause of cancer-related mortality, with 1,089,103 newly diagnosed cases and 768,793 deaths in 2020 (*Sung et al., 2021*). Early detection and curative resection resulted in improved outcomes for patients with early gastric cancer, but the prognosis for the advanced patients remains poor, the five-year survival rate is less than 40% (*Allemani et al., 2018*). The main cause of death in advanced gastric cancer patients is metastasis. In the past two decades, the proportion of patients with metastatic gastric cancer has risen to more than 40% (*Bernards et al., 2013*; *Thomassen et al., 2014*; *Riihimaki et al., 2016*). Thus, it is necessary to study the molecular mechanism of gastric cancer metastasis to elucidate its pathogenesis.

Epithelial-mesenchymal transition (EMT) is an evolutionarily conserved developmental program that is associated with cancer development and confers metastasis properties to cancer cells by enhancing their mobility and invasiveness (*Mittal, 2018*). Cancer cells in primary tumors lose intercellular adhesion mediated by E-cadherin inhibition, break through the basement membrane with increased invasive properties, enter the bloodstream

Corresponding author
Ying Huang, hying_16@163.com

through intravasation, and exit the bloodstream to form micro-metastases (*Lindsey & Langhans, 2014*; *Lambert, Pattabiraman & Weinberg, 2017*). EMT is a cellular process in which epithelial cells lose their epithelial phenotypes and acquire mesenchymal-like characteristics. The initiation of metastasis requires invasion, which is achieved through EMT (*Singh et al., 2018*). Several signaling pathways have been implicated to be involved in EMT, including TGF-$\beta$, Wnt/beta-catenin and Notch, mediating the loss of the epithelial marker E-cadherin and the gain of the expression of the mesenchymal marker N-cadherin and Vimentin (*Gonzalez & Medici, 2014*; *Zhang, Tian & Xing, 2016*; *Brabletz et al., 2018*).

Flotillin-1 is a lipid raft-associated scaffolding protein and it is implicated in a variety of cellular processes including axonal regeneration, cellular adhesion, endocytosis, phagocytosis and signal transduction (*Langhorst, Reuter & Stuermer, 2005*). In addition, Flotillin-1 plays an important role in the progression and development of several malignant carcinomas (*Gauthier-Rouviere et al., 2020*). Upregulation of Flotillin-1 predicts poor prognosis and promotes malignant progression, proliferation and metastasis of multiple tumors, including non-small cell lung, breast cancer, renal cell carcinoma, neuroblastoma, bladder cancer, prostate cancer and hepatocellular carcinoma (*Li et al., 2014a*; *Lin et al., 2011*; *Zhang et al., 2014*; *Tomiyama et al., 2014*; *Guan et al., 2014*; *Jang et al., 2019*; *Zhang et al., 2013*). In particular, studies shown that Flotillin-1 is involved in EMT process of several solid tumors to promote metastasis. In cervical cancer, Flotillin-1 facilitates cell metastasis through Wnt/beta-catenin and NF-$\kappa$B pathway-regulated EMT (*Li et al., 2016*). Flotillin-1 promotes cell invasion and migration by inducing EMT and modulating the cell cycle in lung adenocarcinoma (*Zhao et al., 2018*). However, the detailed characteristics and mechanisms of Flotillin-1 in gastric cancer have rarely been investigated.

In this study, we determined the role of Flotillin-1 in the metastasis process of gastric cancer and molecular mechanisms of Flotillin-1 in the regulation of EMT.

## MATERIALS AND METHODS

### Cell culture

The human gastric cancer cell line SGC-7901 was obtained from GeneChem (Shanghai, China), AGS was acquired from the Shanghai Institute for Biological Science. Both of the cell lines were cultured in RPMI/1640 medium (Gibco, Waltham, MA, USA) supplemented with 10% fetal bovine serum (FBS; Gibco, Waltham, MA, USA) and incubated at 37 °C with 5% $CO_2$. HEK293T cell were purchased from American Type Culture Collection (ATCC) and grown in DMEM medium (Gibco, Waltham, MA, USA) plus with 10% FBS.

### Plasmids and antibodies

Flotillin-1 and Snail were inserted into the pLVX vector. The Flotillin-1 shRNA sequences were cloned into the pSIH vector. The sequences of shRNAs are shown in Table S1.

Antibodies used in this study: anti-Flotillin-1 (#18634; Cell Signaling Technology, Danvers, MA, USA), anti-Snail (#3879; Cell Signaling Technology, Danvers, MA, USA), anti-E-cadherin (#3195; Cell Signaling Technology, Danvers, MA, USA), anti-N-cadherin (#13116; Cell Signaling Technology, Danvers, MA, USA), anti-Vimentin (#5741; Cell Signaling Technology, Danvers, MA, USA), anti-$\beta$-actin (A5441; Sigma-Aldrich, USA),

anti-Flag (#14793; Cell Signaling Technology, Danvers, MA, USA), anti-HA (sc-53516; Santa Cruz Biotechnology, Dallas, TX, USA), Goat anti-Rabbit IgG (AS014; ABclonal, Hubei, China), Goat anti-mouse IgG (AS003; ABclonal, Hubei, China).

## Generation of stable transfected cells

The stable transfected cells including Flotillin-1-overexpressing and -knockdown were generated by lentivirus. The lentivirus was generated using packaging vectors PSPAX2, pMD2G and Flotillin-1-overexpressing plasmid or Flotillin-1-shRNA plasmid. SGC-7901 and AGS cells were infected with indicated lentivirus and performed the stable transfected cell screening using puromycin. The third generation of stable transfected cells were used for subsequent experiments.

## Quantitative real-time PCR (qRT-PCR)

Total RNA was extracted from cells using TRIzol reagent and cDNA was prepared from total RNA using the Quantscript RT Kit following manufacturer's instruction. qRT-PCR assay was performed using SYBR Green Mix following the manufacturer's instruction. The primers used in qRT-PCR were shown in Table S2.

## Immunoblotting assay

Cells were lysed by RIPA lysis buffer plus a cocktail of protease inhibitor. The protein concentration was detected using Bicinchoninic acid (BCA) assay. Total proteins (10 μg for each well) were separated by SDS-PAGE gel and transferred to Polyvinylidene difluoride (PVDF) membranes. The membrane was blocked with 5% fat-free milk for 1 h and incubated with primary antibodies overnight. Next day, the membrane was incubated with conjugated secondary antibody. The protein signal was visualized using ECL detection reagents.

## Migration and invasion assay

Cell migration and invasion assay were performed using 24-well Transwell plate with or without coated Matrigel. Cells ($5 \times 10^4$ for each well) were plated into Boyden chambers with serum-free media. Media containing 10% FBS were added into the bottom chambers, and the cells were cultured for 24 h at 37 °C with 5% $CO_2$. Finally, the cells were fixed by methanol and stained with crystal violet.

## Wound healing assay

Cells ($2 \times 10^5$ for each well) were plated into 6-well plates and scratched using a sterile pipette tip. The cells were washed with phosphate-buffered saline (PBS) and incubated in media containing 2% FBS. Images were obtained at 0, 24, 48, 72 h.

## Immunoprecipitation assay

Cells were treated with 20 μg/ml MG132 for 6 h and lysed with RIPA lysis buffer plus a cocktail of protease inhibitor. For Flag-tagged proteins, cell lysates were incubated with anti-Flag M2 magnetic beads overnight. Finally, the immunoprecipitates were subjected to immunoblotting.

## Ubiquitination assay

For ubiquitination assay, the cells were treated with MG132 for 6 h and lysed using RIPA lysis buffer plus a cocktail of protease inhibitor. The cell lysates were incubated with anti-Flag M2 magnetic beads overnight. The ubiquitin level of Flag-Snail was detected with anti-HA antibody.

## Protein degradation analysis

For Snail protein degradation analysis, cells were treated with cycloheximide (CHX, 50 μg/ml) for 1, 2 h. The protein level of Snail was detected using immunoblotting assay.

## Statistical analysis

Survival analysis was analyzed using a two-sided log-rank test, others were analyzed using unpaired $t$-test. Except for special instructions, all results were shown as mean $\pm$ SEM of three independent experiments. All statistical analyses were performed with Graphpad Prism. A $p < 0.05$ was considered significant.

# RESULTS

## Increased Flotillin-1 predicts a poor prognosis of gastric cancer patients

To identify the effect of Flotillin-1 expression level on the clinical prognosis of gastric cancer, we analyzed the data from The Cancer Genome Atlas (TCGA). A Flotillin-1 amplification occurs in several solid tumors including gastric cancer (Fig. 1A). Compared to normal tissues, the expression level of Flotillin-1 was significantly higher in gastric cancer samples (Fig. 1B). Gastric cancer patients with high Flotillin-1 expression level correlated with a worse survival overall survival, advanced TNM stage and distant metastasis (Figs. 1C–1E). Together, those results shown that Flotillin-1 can participate in the development of gastric cancer.

## Overexpression of Flotillin-1 promotes gastric cancer metastasis

To identify the function of Flotillin-1 in gastric cancer, we overexpressed Flotillin-1 in two gastric cancer cell lines SGC-7901 and AGS. The overexpression efficiency was confirmed by qRT-PCR and immunoblotting (Figs. 2A–2D). To examine the role of Flotillin-1 in gastric cancer metastasis, we performed Transwell and wound healing assays to detect the abilities of migration and invasion. Overexpression of Flotillin-1 not only promoted the migration and invasion of SGC-7901 cells, but also increased the wound healing ability of gastric cancer cells (Figs. 2E–2F). In addition, we obtained the same results in AGS cells (Figs. 2G–2H). Taken together, the results show that overexpression of Flotillin-1 promotes gastric cancer cell migration and invasion and suggested that Flotillin-1 might have a role in cancer metastasis.

## Knockdown of Flotillin-1 inhibits gastric cancer metastasis

To further study the role of Flotillin-1 in gastric cancer metastasis, we knocked down Flotillin-1 in SGC-7901 and AGS cells using shRNA. The knockdown efficiency was confirmed by qRT-PCR and immunoblotting (Figs. 3A–3D). We performed Transwell

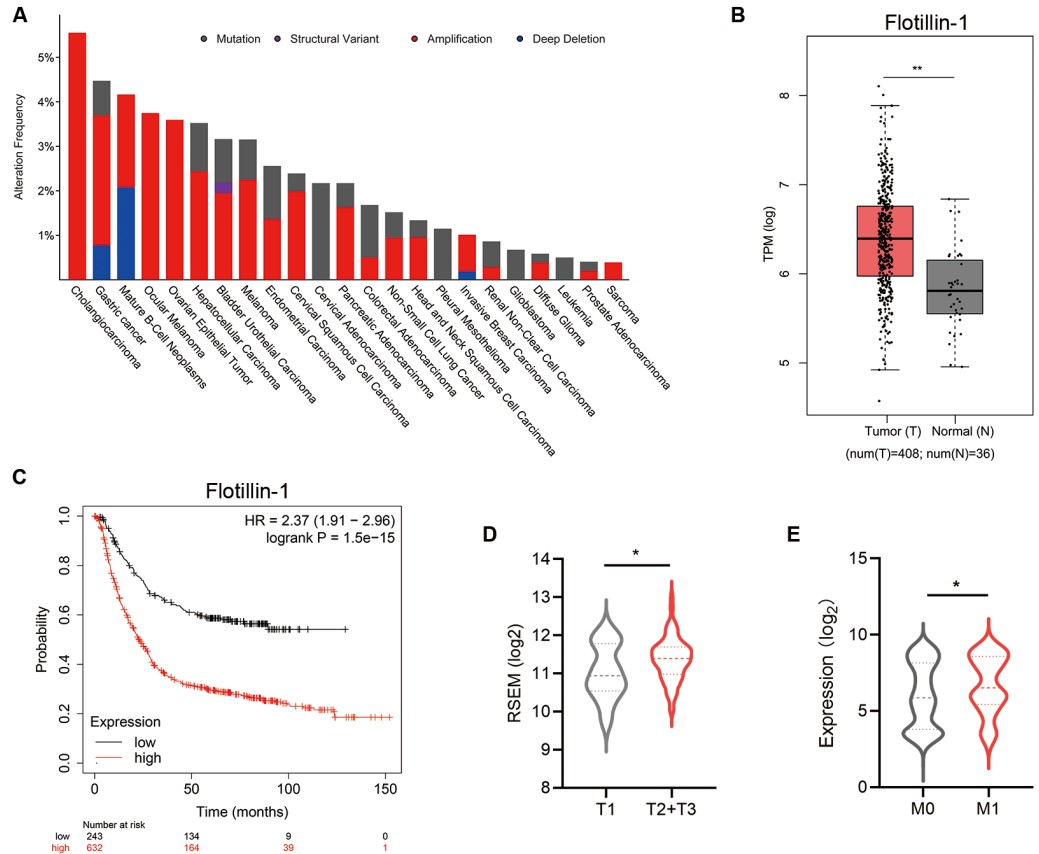

**Figure 1 Increased Flotillin-1 predicts a poor prognosis of gastric cancer patients.** (A) A histogram shows the amplification frequency of Flotillin in 23 tumor types. The data were taken from TCGA Pan-Cancer Altas studies. (B) The expression of Flotillin-1 in gastric cancer tissues and normal samples. T represents tumor tissues ($n = 408$), N represents normal tissues ($n = 36$). (C) Kaplan-Meier plot showing the overall survival of 875 gastric cancer patients stratified by Flotillin-1 expression level. (D–E) Comparison between the proportion of gastric cancer patients with different Flotillin-1 expression level in T stage (D) and distant metastasis (E). $^*p < 0.05$, $^{**}p < 0.01$, $p$ value was calculated with $t$ test, except survival analysis was analyzed using a two-sided log-rank test.

and wound healing assays to detect the cell migration and invasion abilities of Flotillin-1-knockdown and control gastric cancer cells. As expected, knockdown of Flotillin-1 inhibited the migration, invasion and wound healing abilities of SGC-7901 and AGS cells (Figs. 3E–3H). Thus, the results show that knockdown of Flotillin-1 inhibits gastric cancer cell migration and invasion and suggested that Flotillin-1 might have a role in cancer metastasis.

## Flotillin-1 promotes gastric cancer metastasis through inducing EMT

The initiation of metastasis requires invasion, which is achieved through EMT. To identify whether Flotillin-1 promotes gastric cancer metastasis through EMT process. We detected the expression of several EMT markers including epithelial marker E-cadherin and mesenchymal markers N-cadherin, Vimentin and Snail. Overexpression of Flotillin-1

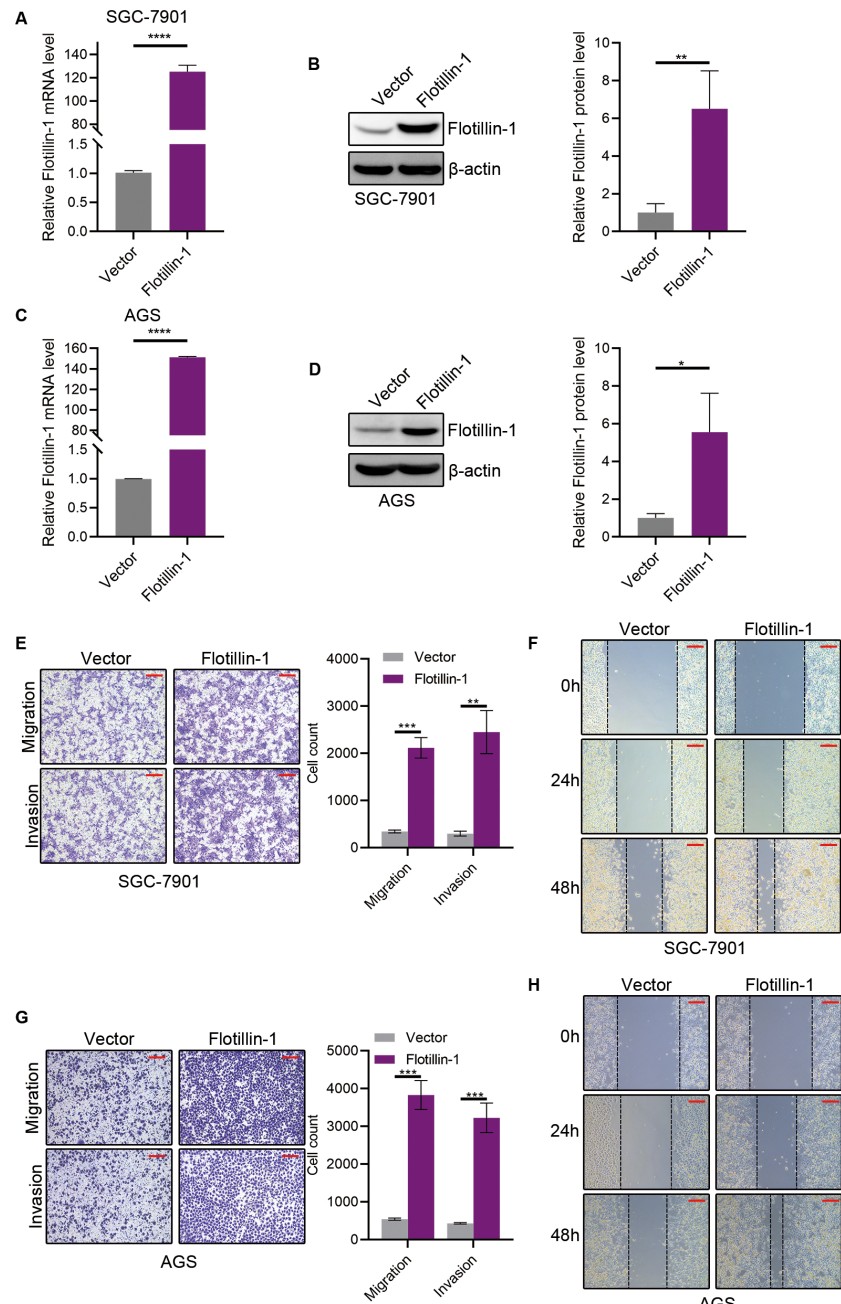

**Figure 2** **Overexpression of Flotillin-1 promotes gastric cancer metastasis.** (A–B) The mRNA (A) and protein (B) levels of Flotillin-1 in SGC-7901 cells overexpressing Flotillin-1. (C–D) The mRNA (C) and protein (D) levels of Flotillin-1 in AGS cells overexpressing Flotillin-1. (E) The migration and invasion abilities of SGC-7901 cells overexpressing Flotillin-1 were detected by Transwell assay. Scale bars, 500 μm. (F) The migration ability of SGC-7901 cells overexpressing Flotillin-1 was examined by wound healing assay. Scale bars, 500 μm. (G) The migration and invasion abilities of AGS cells overexpressing Flotillin-1 were detected by Transwell assay. Scale bars, 500 μm. (H) The migration ability of SGC-7901 cells overexpressing Flotillin-1 was examined by wound healing assay. Scale bars, 500 μm. $**p < 0.01$, $***p < 0.001$, $****p < 0.0001$, $p$ value was calculated with $t$ test, compared to control group. All results were shown as mean ± SEM of three independent experiments.

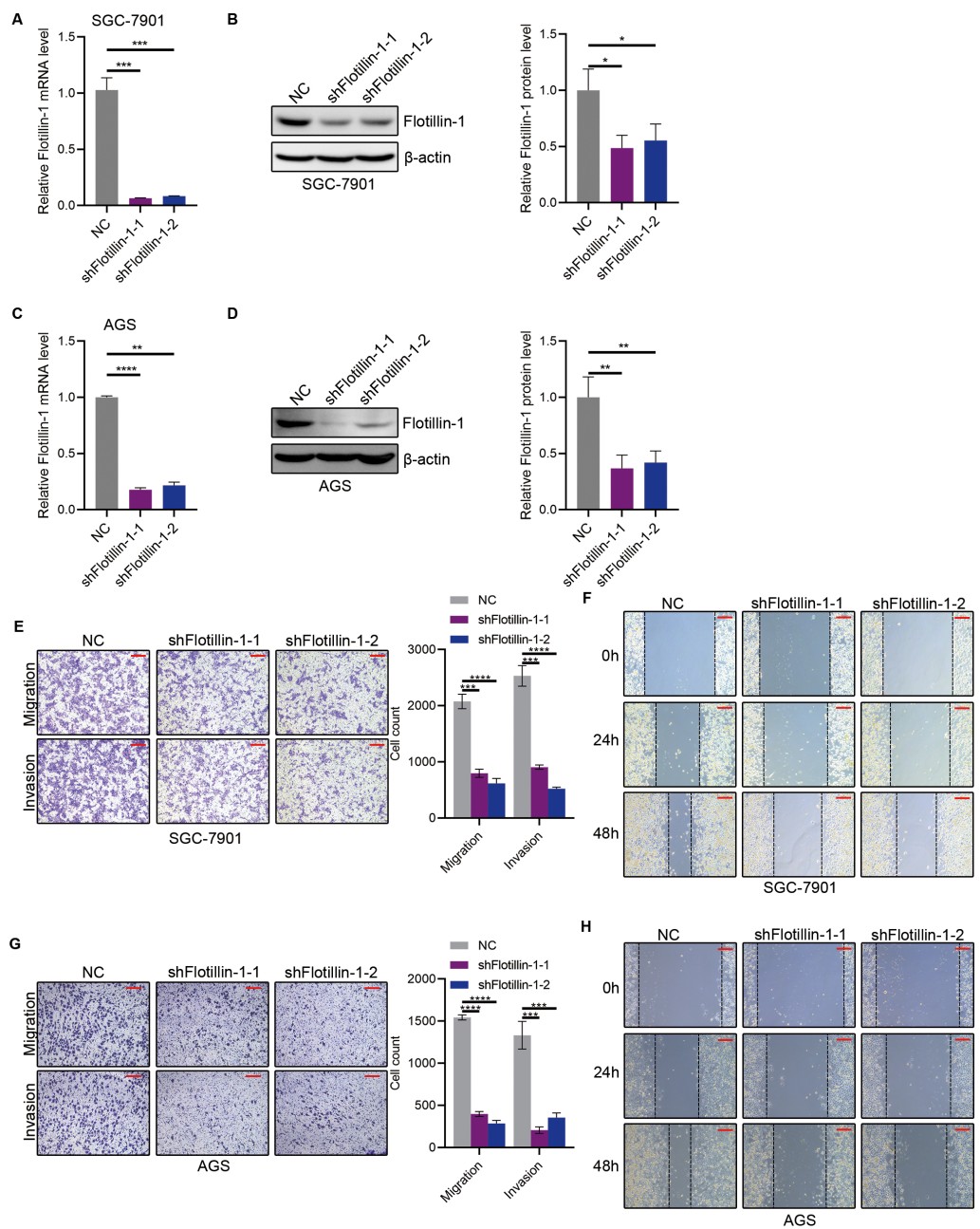

**Figure 3  Knockdown of Flotillin-1 inhibits gastric cancer metastasis.** (A–B) The mRNA (A) and protein (B) levels of Flotillin-1 in Flotillin-1-knockdown (using shRNA targeting Flotillin-1 including shFlotillin-1-1 and shFlotillin-1-2) SGC-7901 cells. (C–D) The mRNA (left) and protein (right) levels of Flotillin-1 in Flotillin-1-knockdown AGS cells. (E) The migration and invasion abilities of Flotillin-1-knockdown SGC-7901 cells were detected by Transwell assay. Scale bars, 500 μm. (F) The migration ability of Flotillin-1-knockdown SGC-7901 cells was examined by wound healing assay. Scale bars, 500 μm. (G) The migration and invasion abilities of Flotillin-1-knockdown AGS cells were detected by Transwell assay. Scale bars, 500 μm. (H) The migration ability of Flotillin-1-knockdown AGS cells was examined by wound healing assay. Scale bars, 500 μm. **p < 0.01, ***p < 0.001, ****p < 0.0001, p value was calculated with t test, compared to control group. All results were shown as mean ± SEM of three independent experiments.

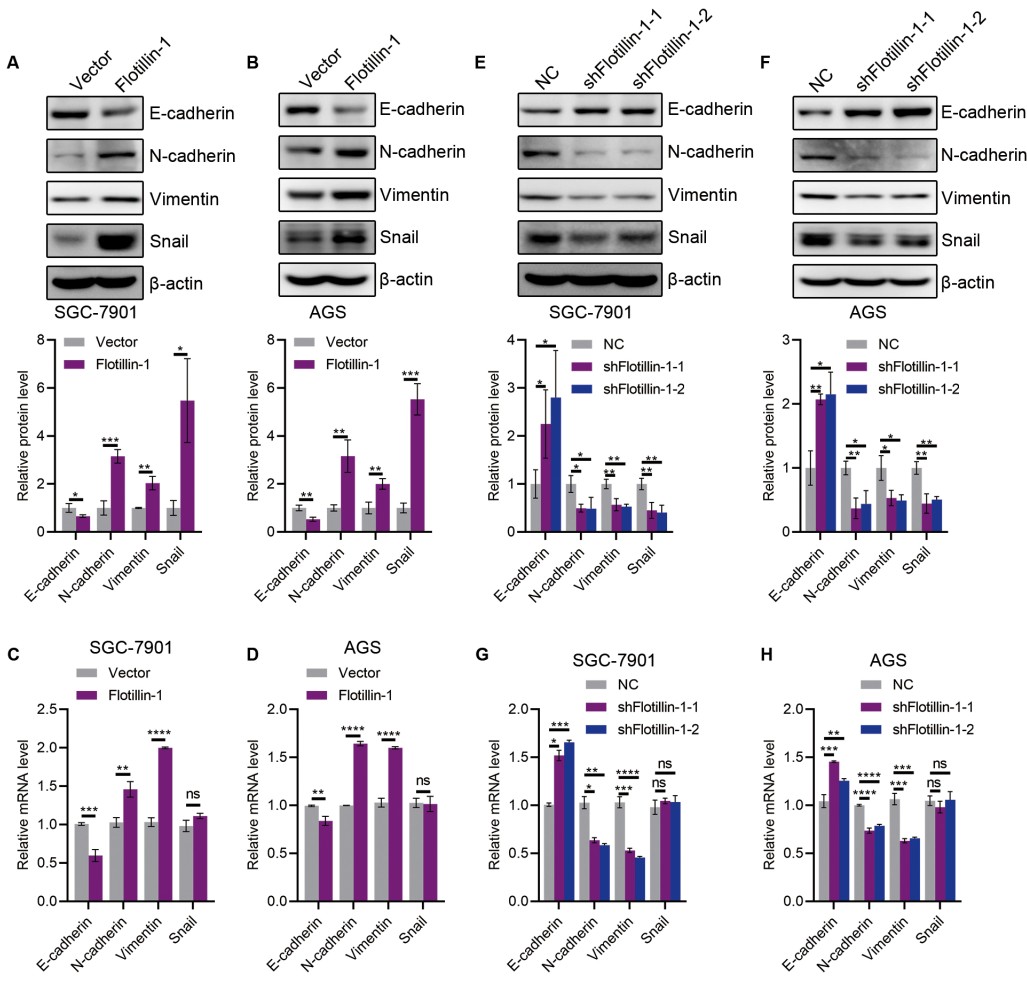

**Figure 4** **Flotillin-1 promotes gastric cancer metastasis through inducing EMT.** (A–B) The protein expression levels of E-cadherin, N-cadherin, Vimentin and Snail in Flotillin-1 overexpressing SGC-7901 (A) and AGS (B) cells. (C–D) The mRNA levels of E-cadherin, N-cadherin, Vimentin and Snail in Flotillin-1 overexpressing SGC-7901 (C) and AGS (D) cells. (E–F) The protein expression levels of E-cadherin, N-cadherin, Vimentin and Snail in Flotillin-1-knockdown SGC-7901 (E) and AGS (F) cells. (G–H) The mRNA levels of E-cadherin, N-cadherin, Vimentin and Snail in Flotillin-1-knockdown SGC-7901 (G) and AGS (H) cells. $*p < 0.05$, $**p < 0.01$, $***p < 0.001$, $****p < 0.0001$, $p$ value was calculated with $t$ test, compared to control group. All results were shown as mean ± SEM of three independent experiments.

downregulated the protein and mRNA levels of E-cadherin, and upregulated the N-cadherin, Vimentin and Snail levels (Figs. 4A–4D). In contrast, knockdown of Flotillin-1 increased the expression of E-cadherin and inhibited the expression of N-cadherin, Vimentin and Snail (Figs. 4E–4H). Moreover, we found that Flotillin-1 regulated the protein level of Snail, but has no effect on the mRNA level. Taken together, those results determined that Flotillin-1 promotes the EMT process in gastric cancer cells.
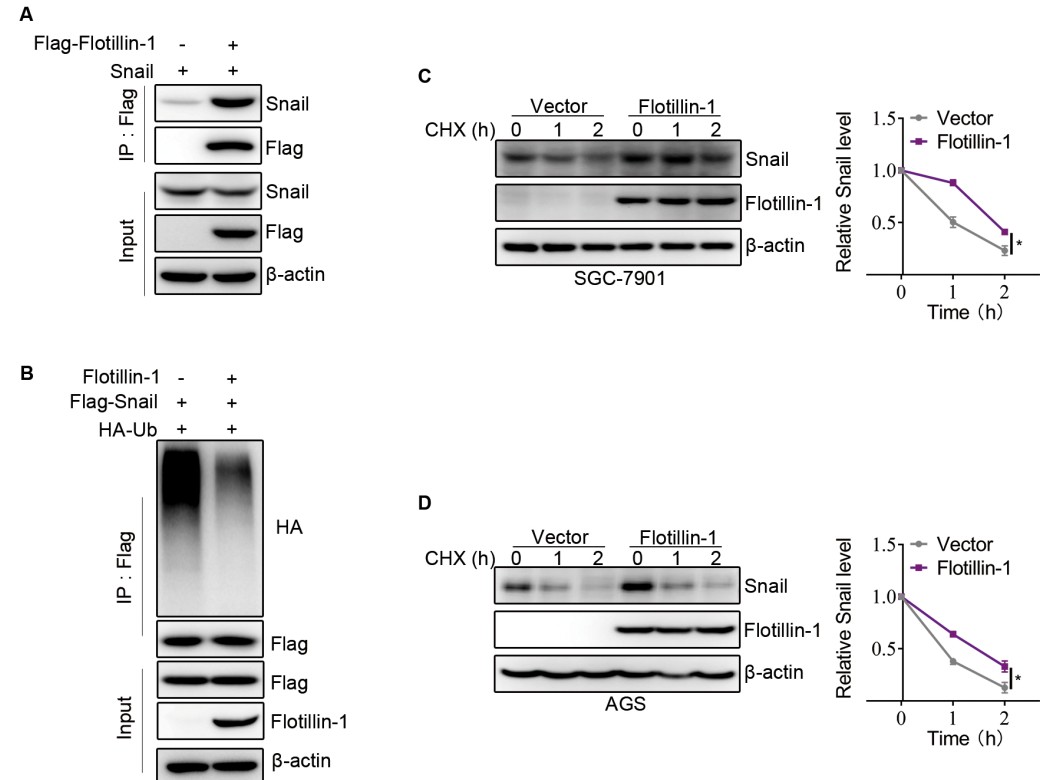

**Figure 5  Flotillin-1 promotes EMT of gastric cancer *via* stabilizing Snail.** (A) Snail and Flag-Flotillin-1 were co-transfected into HEK293T cells. The cells were treated with MG132 for six hours and subjected into immunoprecipation assay with anti-Flag antibody to pulldown Flag-Flotillin-1 immunoprecipitates. (B) HA-ubiquitin, Flag-Snail and Flotillin-1 were co-transfected into HEK293T cells. After MG132 treatment, the cells were subjected into ubiquitination assay with anti-Flag antibody. The ubiquitination level of Snail was detected using anti-HA antibody. (C) Flotillin-1-knockdown and control SGC-7901 cells were treated with CHX as indicated. The degradation of Snail was detected by immunoblotting. (D) Flotillin-1-knockdown and control AGS cells were treated with CHX as indicated. The degradation of Snail was detected by immunoblotting. *$p < 0.05$, $p$ value was calculated with $t$ test, compared to control group at 2 h point.

## Flotillin-1 promotes EMT of gastric cancer *via* stabilizing Snail

Snail, as an EMT-transcription factor, has been confirmed to play an integral role throughout EMT of all types. In addition, Snail is a labile protein and degraded by ubiquitin-proteasome system, and Flotillin-1 regulates the protein expression of Snail but no effect on mRNA level. Thus, we speculated that Flotillin-1 promoted the EMT process through increasing the protein stability of Snail. To identify the hypothesis, we performed immunoprecipitation and ubiquitination assays. As expected, Flotillin-1 interacted with Snail (Fig. 5A) and decreased the ubiquitination level of Snail (Fig. 5B). In addition, Flotillin-1 inhibited the degradation of Snail in gastric cancer cells (Figs. 5C–5D). Taken together, the results shown that Flotillin-1 decreased the ubiquitination level of Snail and inhibited its degradation.

## DISCUSSION

Flotillin-1, a lipid raft protein, is involved in cell migration and invasion, cell signaling, proliferation, differentiation and endocytosis (*Langhorst, Reuter & Stuermer, 2005*; *Affentranger et al., 2011*; *Guillaume et al., 2013*; *Otto & Nichols, 2011*). More studies have shown that Flotillin-1 plays an important role in the development of malignant tumors and may act as a prognosis factor in solid carcinomas (*Li et al., 2014b*; *Liu et al., 2018*). The upregulated expression of Flotillin-1 was associated with tumor cell progression and poor prognosis in hepatocellular carcinoma (*Zhang et al., 2013*). Knockdown of Flotillin-1 impairs cell proliferation and tumorigenicity in breast cancer through upregulation of FOXO3a (*Lin et al., 2011*). MiR-214-3p inhibited cell proliferation and metastasis in hepatocellular carcinoma by downregulating Flotillin-1 (*Liu et al., 2019*). In tongue squamous cell cancer, the expression of Flotillin-1 was correlated with pathological stage, depth of invasiveness, lymph node metastasis, recurrence and shorter survival (*Li et al., 2014b*). In this study, we found Flotillin-1 upregulated in gastric cancer, and the high expression of Flotillin-1 correlated with a worse prognosis. In addition, the migration and invasion ability of gastric cancer cells was upregulated by overexpressing Flotillin-1. Knockdown of Flotillin-1 inhibited the cell metastasis in gastric cancer. Therefore, our results demonstrate Flotillin-1 participates in the development of gastric cancer, especially in promoting metastasis.

EMT is a process by which epithelial cells lose their apical-basal polarity and intercellular adhesion properties and transform into invasive mesenchymal cells, all of which provide conditions for the invasion and metastasis of cancer cells (*Taylor, Parvani & Schiemann, 2010*; *Pearlman et al., 2017*; *Brabletz et al., 2021*; *Blackley et al., 2021*). There are numerous inducers of EMT, and the determinant inducers corresponding to different sites of tumors will be the key to the targeted therapy. In this study, we identified that overexpression of Flotillin-1 decreased the epithelial marker E-cadherin and upregulated mesenchymal markers such as N-cadherin, Vimentin and Snail in gastric cancer cells. E-cadherin was upregulated, and N-cadherin, Vimentin and Snail were decreased in Flotillin-1-knockdown gastric cancer cells. Therefore, our study demonstrates that Flotillin-1 can participate in EMT process and promotes gastric cancer metastasis through inducing EMT.

Snail is a zinc-finger transcription factor that regulates EMT during mesoderm and neural crest development (*Wu & Zhou, 2010*). In breast cancer, Snail expression has been shown to be upregulated in recurrent tumors (*Wang et al., 2013*). In addition, it is associated with reduced metastasis and recurrence-free survival. Similar to zinc finger E-box-binding homeobox 1 (ZEB1), Snail represses cadherin 1 (CDH1) transcription by binding to the E-box in the CDH1 promoter (*Serrano-Gomez, Maziveyi & Alahari, 2016*). It also regulates CDH1 expression in concert with histone methyltransferases (HMT) and DNA methyltransferases (DNMTs) (*Skrypek et al., 2017*). Snail interacts directly with the E-cadherin promoter, recruiting histone deacetylase 1 (HDAC1), HDAC2 and co-repressor Sin3A to the CDH1 promoter and silencing their expression through deacetylation of histones H3 and H4 (*Peinado et al., 2004*). Snail comprehensively affects the expression profile of epithelial cells, besides E-cadherin, it also downregulates claudin, occludin, mucin,

vimentin, fibronectin and matrix metalloproteinases (*Ohkubo & Ozawa, 2004*). Thus, the role of Snail is mainly to downregulate epithelial markers and upregulate mesenchymal markers, and the increased expression of matrix metalloproteinases it regulates makes the cells migratory. In addition, Snail is a labile protein and degraded by ubiquitin-proteasome system. In our study, we demonstrate the interaction between Flotillin-1 and Snail. The protein expression of Snail is upregulated by overexpressing Flotillin-1, and overexpression of Flotillin-1 delays the degradation of Snail in gastric cancer cells. The ubiquitination level of Snail is downregulated in gastric cancer cells overexpressing Flotillin-1. Therefore, Flotillin-1 may interact with a deubiquitinase to inhibit the ubiquitination of Snail in gastric cancer to promote EMT process. This is a direction of our further study.

In summary, our study demonstrate Flotillin-1 can participate in the development of gastric cancer, and promotes gastric cancer metastasis. Increased Flotillin-1 predicts a poor prognosis of gastric cancer patients. Flotillin-1 promotes gastric cancer metastasis through inducing EMT, and promotes EMT of gastric cancer *via* stabilizing Snail. Our study provides a rationale and potential target for the treatment of gastric cancer.

## Funding
This study was supported by the Medical Science Research Project of Hebei Province (20220202). The funders had no role in study design, data collection and analysis, decision to publish, or preparation of the manuscript.

## Grant Disclosures
The following grant information was disclosed by the authors:
Medical Science Research Project of Hebei Province: 20220202.

## Competing Interests
The authors declare there are no competing interests.

## Author Contributions
- Ying Huang conceived and designed the experiments, performed the experiments, prepared figures and/or tables, authored or reviewed drafts of the article, and approved the final draft.
- Yun Guo performed the experiments, prepared figures and/or tables, and approved the final draft.
- Yi Xu performed the experiments, authored or reviewed drafts of the article, and approved the final draft.
- Fei Liu analyzed the data, prepared figures and/or tables, and approved the final draft.
- Suli Dai analyzed the data, authored or reviewed drafts of the article, and approved the final draft.

## Data Availability
The raw data are available in the Supplemental Files.

## Supplemental Information

Supplemental information for this article can be found online at http://dx.doi.org/10.7717/peerj.13901#supplemental-information.

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
