# Peer review of "Flotillin-1 promotes EMT of gastric cancer via stabilizing Snail"

_PeerJ, doi:10.7717/peerj.13901_

## Round 0.1 · original submission · Major Revisions

While all three reviewers were largely positive, some of the points, particularly of the first reviewer, will require some extra work and careful attention.

Under 'experimental design', I am concerned about points 1 , 3 and 5 which must be addressed. Point 4 would be useful to address but otherwise can be discussed and point 2 should be straight-forward.

The issues by all reviewers should be addressed in depth.

Finally, I would ask that you make clear on each figure legend the number of biological and technical replicates for all experiments, upload all the primary data and in particular all the immunoblots for all of the. biological replicates. This latter point is essential.

I look forward to seeing the revision.

·

Basic reporting

The overall writing quality of the manuscript is good. Nevertheless, some problems need to be solved :
- Line 40 : replace « protein and implicated » by « protein and it is implicated »
- Line 117-118: correct the sentence. « Together those results shown that Flotillin may play an oncogene in the development of gastric cancer ». May play an oncogene is too strong. « can participate in the development of gastric cancer ».
- Line 303 : Flotillin 1, 1 is missing
- Figure 1B : replace T by tumor and N by normal.
- Figure 1D : explain what are the T stages.
- Figure 3 shRNA and not only sh has to be written on the legend
- Line 143-144 : the sentence has to be modified, « but has no effect on the mRNA level ».
- Line 160 : the sentence has to be modified, with tumoral cell progression.
- Lines 170-171 : the sentence « Therefore, our results demonstrate Flotillin-1 play a key oncogene in the development of gastric cancer, especially in promoting metastasis » has to be modified. « Therefore, our results demonstrate Flotillin-1 participate in the development of gastric cancer, especially in promoting metastasis » is one possibility.
- Lines 172-182 : the quality of the writing is not good and this part must be rewritten.
- Line 183 : « that » is missing after identified.
- Line 184 : there is a mistake « downregulated mesenchymal markers » has to be replace by « upregulated mesenchymal markers ».
- Line 187 : « that » is missing after demonstrates. The sentence « promotes gastric cancer metastasis through inducing EMT » has to be tuned down, the proposition is « and it overexpression could participate in… ».
- Line 208, again oncogene could not be used. The sentence has to be modified.

Experimental design

The experiments are globaly correctly conducted. Nevertheless, some points need to be addressed. They are detailed below:
1- Control cell lines.
- In figures 2, 4 and 5, using control cells that are the parental cells is not correct. Indeed, the cells (for both cell lines) that overexpress flotillin 1 need to be compared to cells that express a control plasmid.
- In figure 3, using control cells that are the parental cells is not correct. Indeed, the authors have to use a cell line expressing a control shRNA.

2- Explain the cell line generation (both the one that overexpress flotillin 1 or shRNA Flotillin 1). I could not find the information in the material and methods section. This has to be explained. I was wondering whether stable cell line expressing flotillin 1 were generated/used or if the overexpression of flotillins was performed by transient expression ?

3- Quantification of the western-blot data.

Any quantification is shown on the western blot. This has to be done on independent experiments and it is required in order to evaluate the reproducibility of the data.

4 - Other EMT-inducing transcription factors
What about the expression of the other EMT-inducing transcription hy the ? Why the authors analyzed Snail and not the other ones such as Zeb1/2 or Slug?

5 - Flotillin 2 analysis
- Figure 2, the level of flotillin 2 is important to monitore by western-blot since these proteins work together. Is the overexpression of flotillin 1 associated to an increase in flotillin 2 ?
- Figure 3, the level of flotillin 2 is important to monitore by western-blot since these proteins work together. Is the knock down of flotillin 1 associated to a decrease in flotillin 2 ?

Validity of the findings

Some modifications are needed:

1 - Quantification of the western-blot data. (already mentioned above in Experimental design)

Any quantification is shown on the western blot. This has to be done on independent experiments and it is required in order to evaluate the reproducibility of the data.

2- Some sentences are to be modified (detailed in 1: Basic reporting). IN particular saying the flotillin 1 is an oncogene is not possible. The authors express flotillin 1 in transformed cell lines harbouring already oncogenic events.

Additional comments

none

Reviewer 2 ·

Basic reporting

Huang et al. provide information on the role of Flotillin-1 in gastric cancer cell lines expanding the knowledge on the involvement of this protein in tumor biology. The manuscript is clearly written, the introduction gives the necessary information, and the references are relevant. The results are interesting. They show how the modification of Flotillin-1 expression (by overexpression or downregulation) affects migration and invasion of gastric cancer cells.They have studied the expression of some markers of EMT and they have found that flotillin-1 regulates the protein expression of Snail without modifying its mRNA level, likely by increasing its stability. Some modifications in the text are necessary before acceptance of the manuscript. The manuscript might be improved giving more information on the experimental conditions, including the number of replicates in the main text. Most of the conclusions are well stated. However, since only "in vitro" experiments have been performed it should be considered to smooth some of the statements (see below). The demonstration of a role of Flotillin-1 in metastasis formation should require injection of Flotillin-overexpressing and control tumor cells and measurement of the appearance of metastasis "in vivo".






.


.

Experimental design

Methods: The secondary antibodies used are not listed in Methods. No information about how many cells were seeded, or how many micrograms of protein were loaded for western blot is provided. Information on how many independent experiments and how many technical replicates in each experiment should be provided in the main text.

Validity of the findings

Most of the conclusions are well stated. However, since only "in vitro" experiments have been performed it should be considered to smooth some of the statements (see below). The demonstration of a role of Flotillin-1 in metastasis formation should require injection of Flotillin-overexpressing and control tumor cells and measurement of the appearance of metastasis "in vivo".

127 Taken together, the results show that overexpression of Flotillin-1 promotes gastric cancer metastasis.
Since only cell culture experiments have been performed, I suggest to change the sentence: the experiments show that overexpression of flotillin promotes cell migration and invasion and suggest that flotillin might have a role in cancer metastasis.
The same applies for 135-136:Thus, the results show that knockdown of Flotillin-1 inhibits gastric cancer metastasis.

Additional comments

Sentence not clear:
118, 170: “shown that Flotillin-1 may play an oncogene in the development of gastric cancer”.

Figure 1B: Please indicate the meaning of TPM
Figure 1C Please indicate the meaning of HR and the meaning of the different black and red numbers under the X axis. Please make clear in the text the provenance of the samples of tumors expressing low-Flotillin and high-Flotillin, used to build the plot.
Figure 1D Please specify the meaning of RSEM in the axis .


148, 201: I do not understand what the Authors mean with “liable protein”. Do they mean labile?
348 Figure 5: " Flotillin-1-knockdown and control SGC-7901 cells were treated with CHX as indicated".
The legend of the figure contradicts the main text. Do the Authors mean Flotillin-1-overexpressing and control SGC-7901…
It is difficult to understand why control cells do not express Flotillin, while cells labeled as “Flotillin”, which are supposed to be knocked-down for the protein, express it.

Abbreviations:
The authors should provide the information of the meaning of the abbreviations used. It should be convenient to indicate that CDH1 is the gene coding for cadherin 1. HMT is abbreviated but it is used only once. The meaning of DNMT, HDAC should also be provided.

Reviewer 3 ·

Basic reporting

The authors reported that flottilin-1 inhibit ubiquitination of Snail in gastric cancer cells to promote EMT.
The fugires are clear and experiment are well done.

I have two minor comments.

1. The authors declared that gastric cancer is included in 23 cancer series, but I didn't find it. Please let me know the data which shows gastric cancer has amplification of flotillin-1.

2. The authors insisted that ubiquitination of Snail was inhibited by flottilin-1. What is a deubiquitinase involved?

Experimental design

No comment.

Validity of the findings

No comment.

---

## Round 0.2 · accepted · Accept

Thank you for addressing the points raised; I am satisfied with your changes and am therefore recommending acceptance.